# ESH: Design and Implementation of an Optimal Hashing Scheme for Persistent Memory

Dereje Regassa [1,*] , Heon Young Yeom [2] and Junseok Hwang [1]

1 Department of Integrated Program of Smart City Global Convergence, Seoul National University, Seoul 08826, Republic of Korea; junhwang@snu.ac.kr
2 Department of Computer Science and Engineering, Seoul National University, Seoul 08826, Republic of Korea; yeom@snu.ac.kr
* Correspondence: dereje@snu.ac.kr

**Abstract:** Recent advancements in memory technology have opened up a wealth of possibilities for innovation in data structures. The emergence of byte-addressable persistent memory (PM) with its impressive capacity and low latency has accelerated the adoption of PM in existing hashing-based indexes. As a result, several new hashing schemes utilizing emulators have been proposed. However, these schemes have proven to be suboptimal, lacking scalability when implemented on real devices. Only a handful of hash table designs have successfully addressed critical properties such as load factor, scalability, efficient memory utilization, and recovery. One of the main challenges in redesigning data structures for an effective hashing scheme in PM is minimizing the overhead associated with dynamic hashing operations in the hash table. To tackle this challenge, this paper introduces ESH, an efficient and scalable hashing scheme that significantly improves memory efficiency, scalability, and overall performance on PM. The ESH scheme maximizes the utilization of the hash table's available space, thus reducing the frequency of full-table rehashing and improving performance. Consequently, this scheme achieves a high load factor while minimizing the need for rehashing. To evaluate the effectiveness of the ESH scheme, we compare it to widely used dynamic hashing schemes employing similar techniques on Intel Optane® DC persistent memory (DCPMM). The experimental results demonstrate that ESH outperforms CCEH and Dash in terms of data insertion performance, exhibiting a 30% improvement over CCEH and a 4% improvement over Dash. Furthermore, ESH improves the lookup operation by nearly 10% compared to Dash, while achieving a load factor of up to 91%, surpassing its competitors.

**Keywords:** persistent memory; dynamic hashing; directory doubling; hashing schemes

## 1. Introduction

The rapid expansion of data centers often pushes hardware utilization to its maximum capacity, necessitating innovations to redefine existing hardware. In recent times, the emergence of persistent memory devices such as the Intel Optane DC persistent memory module (DCPMM) has addressed the limitations in data access speed and storage capacity by providing persistent, fast, and high-capacity storage solutions [1,2]. The Intel Optane DC persistent memory module (DCPMM) offers substantial storage capacity, delivers competitive performance, bridging the gap between DRAM and flash, and exhibits exceptional resilience to system crashes. With access latency akin to DRAM, remarkable durability, and byte-level addressability, this innovative hardware is exceptionally well-suited for handling latency-sensitive transactions within storage systems.

Furthermore, the novel functionalities of this device have led to alterations in the manner in which data can be retained through direct access via load and store instructions. Substantial advancements have been made in single-level persistence-based applications, enabling direct operation and storage of data on PM without reliance on the storage stack in

OLTP [3,4]. However, data consistency and hardware limitations pose challenges for earlier applications that rely on indexing techniques, which were originally tailored for DRAM environments. Therefore, various persistent indexing techniques have been developed and selected to effectively manage data in persistent memory.

The rapid expansion of data centers often pushes hardware utilization to its maximum capacity, necessitating innovations to redefine existing hardware. This has resulted in the emergence of devices like DCPMM that have addressed these limitations. The limitations of hardware and the need for data consistency pose challenges for earlier applications that utilize indexing techniques [3,4] originally designed for the DRAM environment. As a result, various persistent indexing techniques have been developed and selected to effectively store data in persistent memory.

To enable a constant lookup time for metadata operations, numerous file systems such as ZFS [5], GPFS [6], and GFS/GFS2 [7,8] rely on hash-based indexing. Nevertheless, employing hashing in these systems necessitates a predictable data size, fixed entry size, and pre-allocation of hash buckets from the applications.

In comparison to efforts made towards enhancing tree-based indexing structures for persistent memory, there have been relatively fewer initiatives targeting improvements in hash-based indexing structures. Nevertheless, the current indexing methods do not yield the same benefits on persistent memory (PM) due to issues such as uneven read/write performance and challenges in maintaining crash consistency.

Merely adapting existing indexing structures to function on PM without necessary adjustments to align with the new architecture will not yield the anticipated performance enhancements. Consequently, there arises a compelling need to re-engineer index structures specifically tailored for PM environments. The development of high-performance and scalable indexing structures holds paramount importance for storage systems in achieving rapid query processing capabilities. Consequently, there is a need to redesign index structures specifically tailored for PM. The development of high-performance and scalable indexing structures is essential for storage systems to achieve rapid query processing. Key-value stores extensively employ hashing-based indexing structures in various applications [9,10]. To ensure low overheads and cost-efficient indexing operations, several PM indexes like FAST & FAIR [11], NV-Tree [12], WORT [13], and CCEH [14] have been specifically designed. These indexes not only aim to recover correctly in the case of failures or crashes, but also offer proposals for redesigning indexes [3,4,15–17] by adopting different structures specifically tailored for persistent memory. However, many of these proposals primarily rely on emulators to emulate the behavior of persistent memory.

With the advent of persistent memory, various techniques based on trees and hashing have been designed to optimize indexing. These techniques primarily utilize $B^+$ tree-based indexes [11,14,17,18], hash-based indexes. Notably, significant research has been conducted on hash tables to enhance operation speed in in-memory systems' indexing structures. Achieving an efficient lookup time for mapping values to specific keys in hash-based indexes is crucial for ensuring faster access, regardless of data size.

In this hashing scheme, the allocation of an adequate bucket size for the hash function plays a significant role in determining the buffer cache for the hash table. However, in applications like key-value stores, the sizes of records cannot be predicted due to dynamic insertion or deletion of items. Therefore, a dynamic hashing scheme that involves dynamic resizing becomes the preferred choice to adjust the table size and accommodate records effectively.

During the search for a record in a large storage portion, PM's read/write latency can result in cache misses [18] and PM accesses. Controlling concurrency during these operations requires careful attention, as it introduces additional read/write operations for locking, leading to increased bandwidth consumption. As data insertion increases, the load factor of the hash table rises, necessitating the expansion of the table. This entails rehashing the table and relocating data to newly created buckets.

Opting for a dynamic hashing scheme that incorporates dynamic resizing proves to be the optimal approach for appropriately allocating bucket sizes. This is crucial for ensuring the hash function effectively determines the buffer cache of the hashing table, as rehashing is an expensive operation that involves doubling the number of buckets incrementally, determining the new index for values to be mapped to their new locations. The movement of existing records to new bucket locations degrades throughput and temporarily halts access to the indexes during rehashing. Additionally, the read/write latency of PM further contributes to the overall query latency. This challenge is not adequately addressed by the existing research, yet it needs to be addressed.

In this research paper, we propose a scalable hashing scheme designed specifically for efficient storage of key-value pairs on persistent memory (PM), addressing the limitations as discussed above. Our approach incorporates mechanisms that optimize the utilization of the hash table by fine-tuning the directory entries and segments, allowing for the accommodation of more records within existing buckets. This is achieved by extensively utilizing the segment before resorting to costly full-table rehashing, and this is a new approach that this work is bringing forward.

We introduce an efficient and scalable hashing scheme, referred to as ESH, which possesses the following properties:

(i) It efficiently utilizes the available space in the hash table by redistributing overflow records from a bucket to neighboring buckets within the segment.
(ii) It reduces or delays the need for full-table rehashing, resulting in improved insertion performance.
(iii) It increases the load factor and enhances PM/memory utilization efficiency without significant performance loss, even for varying data sizes and thread counts.
(iv) It presents a scalable hashing scheme that minimizes unnecessary PM read and write operations, conserving PM bandwidth and demonstrating scalability in multi-threaded environments.

The rest of this paper is organized as follows. Section 2 describes the background and motivation. Section 3 presents the design and implementation of the proposed idea using a designed approach. Section 4 shows the experimental evaluation and its results. Section 5 discusses the related work, and Section 6 concludes the paper with recommendations for the use of the output on a similar scale.

## 2. Background and Motivation

### 2.1. Optane Persistent Memory

In contemporary computing applications, persistent memory has found utility in various domains such as databases, storage, cloud computing/IoT, and artificial intelligence. Its introduction has brought about a fundamental shift in computing architecture. Notably, Intel's DC persistent memory (PM) has redefined traditional memory architecture by offering significantly larger capacity than DRAM at an affordable price point. It boasts features such as high capacity, low latency, and real-time crash recovery for storage purposes. This memory is directly populated and accessed through the existing memory bus, utilizing CPU load and store instructions, thus avoiding the high overheads associated with conventional interfaces. The increased capacity and improved security through hardware-level encryption also address significant business challenges.

As the capacity limits of volatile memory (DRAM) [19] have been reached, non-volatile persistent memory has emerged as an alternative solution. However, traditional hashing techniques designed for DRAM are inefficient when applied to persistent memory. Therefore, we have devised an efficient hashing scheme that leverages the unique features of persistent memory to overcome these inefficiencies.

### 2.2. Optane Architecture and Instructions Support

To guarantee data persistence, the integrated memory controller (iMC) operates within the asynchronous DRAM refresh (ADR) domain. Intel's ADR feature ensures that CPU

stores reaching the ADR domain will remain intact even in the event of a power failure [20]. In this system, the iMC maintains separate read and write pending queues (RPQs and WPQs) for each Optane DIMM, as depicted in Figure 1a. Once data reach the WPQs, the ADR mechanism ensures their survival during power loss. The actual access to the storage media takes place after address translation. Notably, the Optane controller translates smaller requests into 256-byte accesses of the Optane block, leading to write amplification. As the NUMA configuration is used in this work, address space interleaving (Figure 1b), the effect of the iMC contention will be reduced when the size of the random access is the same or close to the address space interleaving size of 4 KB where the stride size becomes 24 KB for six PM inserted into all DIMMs. Applications and file systems modify the contents of Optane DIMMs by utilizing store instructions, thereby ensuring data persistence. Intel's instruction set architecture (ISA) offers specific instructions for flushing cache lines back to memory, such as *clflush* and *clflushopt*. The *clwb* instruction is utilized to write back the cache lines. Additionally, applications employ the *ntstore* instruction to write directly to memory without involving the cache. To ensure that the data is persistently flushed, applications need to issue the *sfence* instruction, which guarantees the flushing of data to persistent memory.

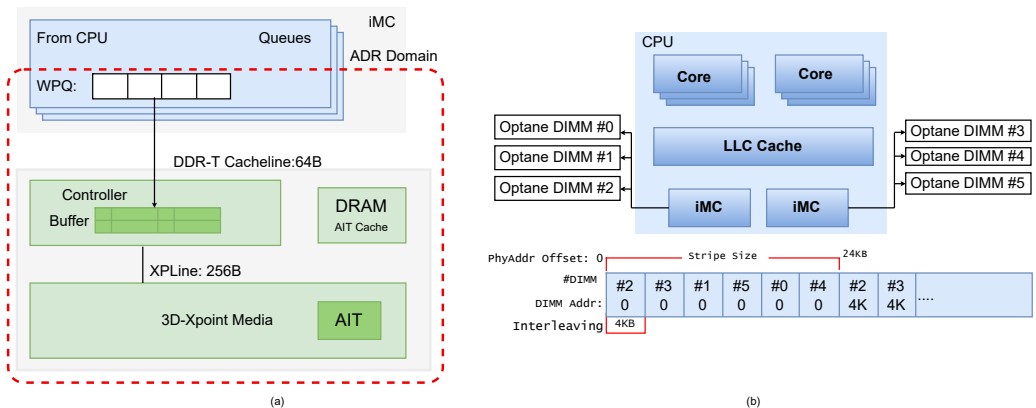

**Figure 1.** Optane architecture: (**a**) overview of Optane DIMMs; and (**b**) Optane DIMMs interleaving strategies.

Developing significantly faster data structures for persistent memory necessitates programming changes that can introduce complexity and increase the likelihood of errors [21]. Nevertheless, it remains crucial to consistently store data in the order of the store operation. Standard approaches are typically employed to ensure this consistent storage of data. Consequently, the design of a write-optimized and scalable hash table specifically tailored for persistent memory incorporates these features to achieve the desired performance improvements.

### 2.3. Dynamic Hashing

Hashing is a technique that has long been employed in memory-based data storage. Dynamic hashing, in particular, is commonly used to reduce the length of string characters. This scheme operates by dynamically adjusting, rearranging, and resizing the characters to align with the data access patterns, resulting in faster and more efficient data storage.

There are various dynamic hashing schemes available that utilize in-place updates for read–modify–write operations. Additionally, some schemes incorporate optimized Cuckoo hashing techniques to minimize locking for smaller key–value stores [22,23]. Moreover, there are similar schemes like Dash [18], which have also implemented linear hashing. However, it has been observed that excessive reads occur during end-to-end operations due to the requirement of a linear scan. As a result, in this paper, we have chosen a scheme that not only increases the hash table as the data size increases but also allows for dynamic modification of the hash function to accommodate the growth or shrinking of the

records. Therefore, we have opted for an extendible hashing scheme that possesses this unique capability.

**Extendible hashing**: This is a dynamic hashing technique that performs incremental rehashing operations, minimizing the impact of hash table growth on applications when compared to standard full-table rehashing. As the size of the in-memory data grows, rehashing traditional hash tables introduces higher latency, necessitating the utilization of improved rehashing techniques for persistent memory. Various methods, such as linear probing, separate chaining, Cuckoo hashing, CCEH [14], and Dash [18], are employed to mitigate the overhead during rehashing. CCEH, a variant of extendible hashing, is specifically designed to optimize the access of hash table buckets for cache-line access, significantly reducing the number of cache-line accesses required. Additionally, CCEH minimizes the overhead in directory operations by grouping a number of buckets into intermediate-sized segments. This approach reduces the size of the directory to cache-line-sized buckets and simplifies rehashing management during failure recovery.

Extendible hashing is a dynamic hashing scheme that employs a directory, known as a bucket address table, to locate specific queries and retrieve records with a given key. The hashing function used in this scheme is adaptable and can be dynamically altered to effectively manage directories and buckets for data hashing. The records themselves are stored in buckets, and these buckets are pointed to by the global directory. The directory contains bits that determine the directory entry, represented in binary form, which then points to the respective buckets. Figure 2 illustrates the arrangement of directory entries and their connections to the buckets, where each bucket stores key–value pairs.

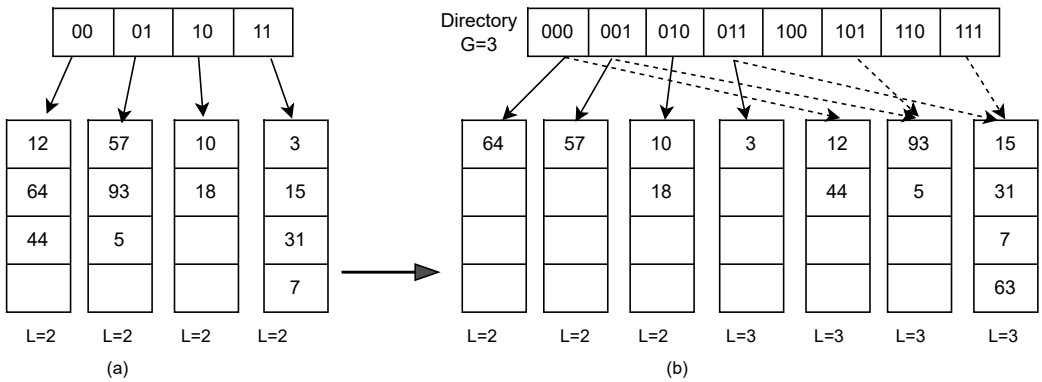

**Figure 2.** Extendible hashing and directory doubling: (**a**) before directory doubling; and (**b**) shows depth increase after doubling.

In extendible hashing schemes, the maximum number of buckets is represented as $2^G$, where G denotes the global depth. Lookup and update operations navigate through the hash table by following the corresponding directory entry pointer, leading to the desired buckets. Consequently, the hash table can accommodate a maximum of $2^G$ directory entries, each pointing to a specific bucket. Additionally, buckets are tracked using the local depth (L), which aids in determining when a bucket becomes full and initiates actions such as bucket splitting or directory doubling.

In the depicted scenario, as illustrated in Figure 2a, when a record (e.g., 63) is inserted into the hash table and the last bucket is already full, the process of directory doubling is triggered due to the local depth being equal to the global depth (L = G). In Figure 2b, the directory entry is doubled, resulting in an increase in the local depth of the overflowing bucket. Subsequently, the records are rehashed and inserted into the newly expanded hash table.

In situations where multiple directory entries point to a single bucket and the local depth is smaller than the global depth, it signifies an overflow condition. When bucket overflow occurs, the process of bucket splitting is initiated, resulting in the division of the overflowing bucket into two separate buckets. The execution of this operation depends

on whether the local depth of the overflowing bucket is equivalent to the global depth prior to the split. It is important to note that this process may or may not involve directory doubling. During the splitting process, the local depth for the resulting buckets is increased. If the local depth becomes equal to the global depth, an overflow condition occurs, leading to directory doubling. Directory doubling is necessary when the available $G$ bits are insufficient to differentiate the search value of the overflowing bucket.

**Extendible hashing on PM**: Previous studies have explored the adaptation of extendible hashing for use on PM [14,18]. These efforts have focused on minimizing PM access to enhance the efficiency of the hashing scheme. One approach involves grouping buckets within a segment and utilizing a single directory entry for these buckets, which effectively slows down the growth of the directory [14]. Additionally, Dash [18] has made attempts to decrease unnecessary read and write operations, aiming to conserve bandwidth and mitigate the impact of high end-to-end read latency.

### 2.4. Effect of NUMA Access

Modern main memory systems consist of dual inline memory modules (DIMMs) that contain DRAM chips. These DIMMs are combined to form the total main memory capacity of the system. As parallelism continues to increase in multicore and multi-processor nodes within clusters, distributed memory banks and buses with varying access costs have become prevalent. This approach, known as non-uniform access (NUMA), assigns separate memory banks to each processor, distributing memory and CPUs across different nodes. This mitigates the performance impact caused by multiple processors or cores accessing the same memory simultaneously. Processors within the same node have faster access to their local memory while accessing remote memories on other nodes incurs higher latency. This distinction in memory access speed contributes to overall performance improvements.

Today, high-performance servers commonly utilize non-uniform memory access (NUMA) architectures, which feature intricate memory hierarchies. This adoption of NUMA by computer architects allows for accommodating numerous cores within a single computer. The cores are organized into nodes, with each node sharing the last-level cache (LLC) and memory. However, traditional assumptions regarding memory, such as access time and memory stall, no longer hold true in these machines. NUMA machines often experience longer memory stalls, and designing accurate data structures to optimize performance becomes more challenging in such environments [24].

Therefore, in order to achieve high performance, it is essential to design indexing structures that effectively leverage the benefits of NUMA. These designs need to prioritize efficient time complexity, minimize the synchronization overhead, and ensure cache-efficient memory access patterns. The latency and bandwidth in NUMA systems vary based on the nodes involved in data access and storage. Extensive research has been conducted on NUMA optimizations, which primarily revolves around strategically placing threads and data across nodes to minimize latency and maximize bandwidth [25–29]. The performance of a system is influenced by both hardware and software design choices that take into account the memory system architecture. In multicore processors, for instance, the cores share on-chip memory systems' resources such as memory controllers, last-level caches, and pre-fetcher units. These shared resources play a crucial role in determining the overall system performance. When there is competition for the utilization of these resources, it can result in a decline in performance [30,31]. The operating system scheduler plays a vital role in mitigating contention issues related to shared last-level caches. Addressing problems associated with data locality can be achieved through either profile-based or dynamic memory migration techniques [30]. The performance of non-volatile file systems can be significantly impacted by the challenges of accessing non-local memory (NUMA effects) [32]. Introducing NUMA-aware interfaces to file systems that utilize non-volatile memory modules can alleviate these issues. There is ongoing research focused on enhancing NUMA locality, with some proposing the redesign of data structures to incorporate NUMA awareness, allowing for optimal utilization of the internal features of these structures [33–36].

Optimizing memory allocation in NUMA systems is crucial for maximizing system performance. Furthermore, the selection of efficient NUMA-aware file systems significantly impacts overall performance [37]. With the introduction of Optane DIMMs, there is a growing body of research focused on redesigning storage systems to fully leverage their potential [16,38].

### 2.5. Inter-Thread Interference

The concurrent execution of multiple threads is a crucial aspect that researchers focus on to achieve high performance in main-memory database systems utilizing multicore architectures. To achieve efficient concurrency on modern CPUs, latch-free (lock-free) indexing structures are implemented to avoid bottlenecks. For instance, applications like MemSQL utilize lock-free skip lists [39], while Microsoft's Hekaton main-memory OLTP engine employs lock-free B+Trees [40]. Designing lock-free algorithms for indexing structures can be complex, relying on atomic CPU hardware primitives like compare-and-swap (CAS) instructions to atomically modify the index state. Unlike lock-free approaches, extendible hashing schemes typically require multiple word updates during directory doubling and segment splitting, making them more susceptible to exposing intermediate states to other threads. This complexity is amplified in multicore systems and may result in race conditions if intermediate states are accessible by other threads.

### 2.6. Locking

Optimizing existing hashing-based indexes for persistent memory (PM) primarily focuses on crash consistency and write optimizations. However, there has been relatively less emphasis on studying the impact of blocking during the hashing process [14–16]. Using a global lock to block access to the entire hash table is not suitable for time-sensitive applications, as it hinders concurrent access from other threads. Instead, it is more favorable to implement a well-designed fine-grained locking scheme that protects a limited number of buckets or even a single bucket. In this regard, the efficient and scalable hashing (ESH) scheme utilizes optimistic locking at the bucket level. This reduces the blocking of table blocks from other insert processes. Additionally, the lookup operations in this scheme are lock-free, allowing multiple processes to access the same record in a bucket.

### 2.7. Impact of Segment Resizing

A common use of a hash table is to store key–value pairs, enabling associations between keys and values for dictionary implementation or determining if a key is part of a set of keys for set operations. When the initial capacity of the hash table is insufficient for the number of items to be inserted, hash table resizing becomes necessary. This resizing process incurs an overhead and affects the insertion throughput. The efficiency of hash tables relies on performing resizing operations at the lowest possible cost.

If the bucket size is small, the hash table fills up quickly, triggering the hash function to initiate the costly rehashing operation, which impacts performance. Allocating a relatively larger hash table proves advantageous as it delays the need for resizing. In our scheme, intermediate resizing occurs during segment splitting to expand the space before a full-table rehashing is required. Segment splitting involves significantly fewer accesses to persistent memory compared to rehashing the entire table.

In the ESH scheme, a bucket size of 256 bytes is used, allowing for the storage of more records once the hash table is created by the hash function. When this bucket becomes full, segment rehashing redistributes the key–value pairs to the buckets after the segment splits. This feature, combined with a larger bucket size, helps reduce the frequency of resizing operations, ultimately contributing to improved performance.

## 3. Related Work

Persistent memory, with its durability, byte-addressable nature, and access latency similar to DRAM, offers significant advantages for building applications that rely on

extensive memory systems. However, the shift in memory architectures has rendered traditional data indexing methods inefficient in terms of data consistency [13]. To address this challenge, several existing studies have focused on enhancing tree-based indexing structures for persistent memory. These advancements aim to achieve reasonable lookup times and improved recovery capabilities [3,11,13,17,41]. With the increase in the size of in-memory data, the process of rehashing traditional hash tables introduces higher latency, necessitating the development of improved rehashing techniques tailored for persistent memory. To mitigate the overhead during rehashing, various techniques such as linear probing, separate chaining, Cuckoo hashing, CCEH [14], and Dash [18] have been employed. CCEH, a variant of extendible hashing, has been specifically designed to optimize the access of hash table buckets by enabling cache-line access, thereby minimizing the number of cache-line accesses. Additionally, CCEH reduces the overhead in directory operations by grouping a number of buckets into intermediate sizes known as segments. This approach helps to reduce the size of the directory to cache-line-sized buckets and also streamlines the rehashing management during failure recovery [42–44].

Extendible hashing was initially developed to address the needs of time-sensitive applications that utilize a trie for efficient bucket lookup. It employs a process called rehashing, which is an incremental hierarchical operation aimed at populating the hash table effectively. To achieve this, it is crucial to track the dynamic allocation of buckets and their corresponding pointers within the hash table. Several research efforts have been dedicated to enhancing the efficiency of rehashing schemes, given that time-sensitive applications are particularly susceptible to the impact of hash table rehashing compared to the growth of the tables themselves.

The growth of the table size also significantly influences the performance of record lookups. Since extendible hashing employs a directory to index dynamically added or removed buckets during runtime, the process involves splitting existing buckets and creating new ones with reorganized values. This expansion of the directory is necessary to accommodate additional storage for pointers to the new buckets. Typically, this can be achieved linearly by organizing the buckets using directory entries that point to individual buckets. Designing an appropriate splitting and rehashing strategy tailored to different workloads is a fundamental approach to achieving optimal in-memory systems [45,46].

Cuckoo-based hashing, as explored in [47], offers a way to reduce write operations to PCM (phase change memory) while maintaining higher memory efficiency. It achieves this by displacing randomly chosen records to alternative buckets [47]. The insertion of records into buckets is carried out using independent hash functions. However, it is important to note that the performance of this approach is still slower compared to linear probing and Cuckoo hashing [41,48,49]. To further improve the lookup cost, binary trees have been employed to achieve logarithmic scale efficiency [50]. Additionally, these binary trees have been divided into two-level hashes for enhanced performance [15].

The primary focus of many research studies is to reduce the cost associated with full-table rehashing and improve the load factor of hash tables. Some proposals have explored the idea of incorporating additional levels [47] to store records that are actively maintained in memory rather than being directly stored in the hash table. Another notable approach gaining popularity is the use of cache-line-level indexing with a failure atomic structure, as described in [14]. This technique dynamically manages hash expansion on persistent memory (PM) and guarantees a constant lookup time for the hash table. By setting the size of each bucket to match a cache line, the number of cache-line accesses is minimized, resulting in reduced overhead when accessing data spanning multiple cache lines.

An alternative approach, as described in [18], utilizes a PM tree structure to avoid unnecessary reads during record probing. This approach incorporates fingerprinting techniques from PM trees and introduces a lightweight one-byte hash to detect keys efficiently, thereby conserving PM read/write bandwidth. The strategy employed by this approach involves postponing segment splits, which leads to improved space utilization. The implementation of this approach for PM utilizes the PMDK libraries [51].

A comprehensive study of main-memory hash joins in storage class memory (SCM) and evaluation of two state-of-the-art join algorithms is studied in [52]. It explores the design space, identifies the most crucial factors to implement an SCM-friendly join, and compares the performance of partitioned hash join (PHJ) and non-partitioned hash join (NPHJ) in real SCM environments. It provides valuable insights into the design and evaluation of main-memory hash joins in SCM, and the experimental results can help guide future research in this area.

As the size of the dataset grows, the process of rehashing all the records becomes a challenge and a resource-intensive operation. To address this issue, extendible hashing is specifically designed for applications that prioritize time sensitivity. This hashing technique allows for incremental rehashing operations as the data size increases or when bucket overflow occurs. During the process of rehashing, the goal is to allocate all records to their respective buckets based on their hash keys. In this process, the directory entry pointers in the extendible hashing scheme are doubled as the depth increases, as illustrated in Figure 2. The directory, represented by the bucket address pointer, is responsible for pointing to the hash buckets [14]. As the size of the hash table increases, the directory is updated accordingly.

Developing a dynamic and scalable hash table that is compatible with the new architecture of persistent memory hardware, and capable of operating efficiently under high load factors and providing instant recovery, is of the utmost importance [15,18,47,53]. Furthermore, the design of hashing schemes that minimize the overhead associated with dynamic memory management, while improving hash table lookup time and other related operations, is equally crucial. This work aims to contribute to these ongoing efforts.

## 4. Design and Implementation

Like other approaches [14,18], ESH is designed using a two-layer segmentation of buckets. In this paper, we introduce an efficient and scalable hashing scheme that optimally stores key–value pairs on persistent memory. It achieves higher memory utilization by effectively utilizing the available free space within the buckets of the same segment, allowing for the storage of more records in each segment.

### 4.1. High-Level Design

The proposed scheme is specifically designed to take advantage of the performance characteristics offered by Optane persistent memory. By employing advanced hashing techniques, it aims to enhance performance and scalability while addressing challenges commonly encountered in hashing index structures. As such, this work focuses on several key design principles, including:

(a) **Avoid both unnecessary reads and writes**: In a hashing scheme, write operations typically involve frequent access to the underlying storage media, which significantly impacts the overall performance of the scheme. These frequent read and write operations have a cumulative effect on all operations, including reads, writes, and other related tasks. Furthermore, devices with slower speeds compared to DRAM experience even more severe performance overhead due to the frequency of these read-and-write operations. To achieve high end-to-end performance, ESH addresses this issue by reducing unnecessary reads and writes to persistent memory.

(b) **Bucket level locking to allow multi-threading:** A well-designed locking strategy aims to minimize the frequency of lock and unlock requests for sequential data access and manipulation, resulting in reduced CPU costs. In order to achieve better concurrency, ESH adopts a fine-grained locking approach that reduces the need for locks and unlocks, thus minimizing lock contention. Specifically, during write operations, only the bucket being operated on is locked, allowing other threads to access different buckets concurrently. Other operations are lock-free, enabling higher levels of concurrency. While readers can access buckets freely, certain operations such as segment splitting and directory doubling are not lock-free to ensure data consistency. To prevent data

inconsistencies, the active writer thread takes responsibility for creating and locking segments or directories during splitting or doubling. This approach in ESH focuses on using locks at the minimal data block level, specifically, the bucket level.

(c) **Optimistic scaling on multicore machines**: To fully leverage the parallel resources of CPUs, ESH incorporates an optimistic scaling approach in its design. Traditional research efforts have primarily concentrated on minimizing cache-line flushes and utilizing PM writes to achieve scalable performance. However, these approaches encounter scalability challenges when deployed on actual PM devices. Due to the limited bandwidth of PM, ESH focuses on reducing unnecessary PM reads and implementing lightweight concurrency control mechanisms to further minimize PM writes, ensuring persistence with reduced overheads. As a result, ESH treats a bucket as a single block of PM, effectively reducing the PM access overhead.

In Figure 3, we can observe the comprehensive architecture of the EHS. To explain the architecture, we can take the insertion of data to the hashing table. A bucket is divided into two sections. The first section, called metadata, stores information about the bucket, facilitating efficient access and ensuring data consistency. The second section is dedicated to storing the actual records, including key–value pairs. Consequently, each directory entry points to a segment that comprises a predetermined number of regular buckets, along with the necessary metadata for locking, state information, and overflow records from adjacent buckets that cannot accommodate additional insertions.

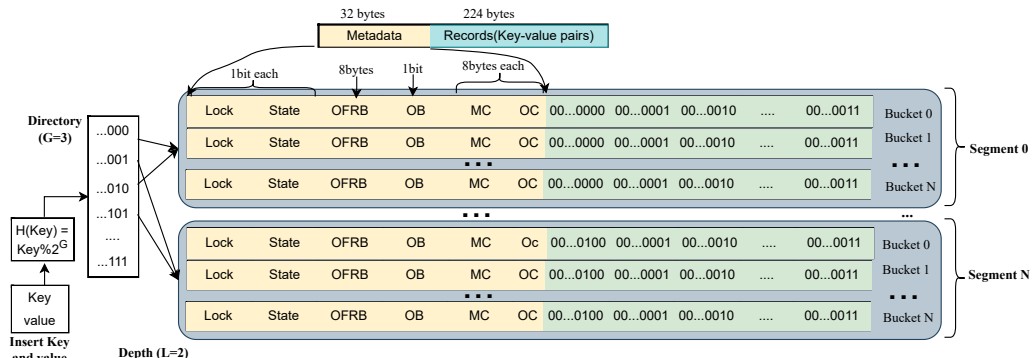

**Figure 3.** Overall architecture of bucket and segment structure for scalable hashing scheme. OFRB: overflow records address byte; OB: overflow bit; MC: membership counter; OC: overflow counter.

To ensure data consistency, the scheme verifies the state of a bucket before performing any operations on it, utilizing the *state* flag present in the metadata. If a bucket is found to be in an inconsistent state due to a previous operation that has not yet been persisted (e.g., due to a crash or power failure), no further operations are carried out on the affected bucket, and the data within it is considered invalid. To avoid the costly logging of all operations, a successful insertion is only acknowledged after the metadata has been persisted. Otherwise, if the metadata persistence fails, the record is discarded. In Figure 3, the last four bits of the hash function's result play a critical role in determining the segment and bucket for record insertion. These bits serve as directory entries, allocated to segments by the hashing scheme. The metadata provides additional information to specify the exact bucket for record storage. Further details regarding the metadata are described in the subsequent section.

**Metadata**: The metadata section in each bucket holds crucial information, such as the bucket's status, lock state, and overflow indicators, enabling fast access to the bucket. This valuable information, as shown in Figure 3, is essential for performing various operations. In the implementation, the metadata occupies 32 bytes of the total bucket size, as required for conducting the experiment.

**Locking (lock)**: Locking ensures that a process remains in memory and imposes restrictions on access to a resource or bucket when multiple threads are executing. In real-

time environments, it is crucial to guarantee the locking of a process in memory to minimize latency for data access, instruction fetches, buffer passing, and other operations. It serves the purpose of enforcing mutual exclusion and concurrency control. In the proposed scheme, the lock indicator is a metadata component that stores locking information for writer threads. Writer threads are required to acquire the lock by setting the lock bit to 1 before writing to the bucket, ensuring consistency. Once the first thread releases the lock and sets the lock bit back to 0, other threads can attempt to access the same bucket again.

**State**: The state parameter is a Boolean flag that determines whether the system has been shut down properly. In certain scenarios, the hashing scheme may be left in an inconsistent state due to unexpected events such as power failure. When a process completes its operation on a bucket, it updates the state to 0, indicating that the flag is correct. When other threads access the same bucket, they check the state parameter. If the state is set to 1, it signifies that the prior process did not close the system correctly. In such cases, the data should be discarded or a recovery process should be initiated to restore consistency.

**Overflow bit (OB)**: The overflow bit is a Boolean indicator used to determine the status of a bucket during insertion. It helps to identify whether the bucket is full or not.

**Overflow records address byte (OFRB)**: In ESH, OFRB is used to store the address of a neighboring bucket where records are moved when the initial bucket becomes full. Instead of immediately triggering segment splitting or directory doubling, the scheme checks for available space in neighboring buckets within the same segment. By storing the record in a neighboring bucket, the OFRB holds the address of the bucket to which the record has been moved. This allows for easier search operations as it indicates the address of the bucket holding the record.

**Overflow counter (OC)**: In ESH, OC is used to differentiate between records that are originally hashed to a specific bucket and those that have been moved from neighboring buckets due to overflow. This differentiation is important to avoid confusion during search operations. When searching for records, the overflow counter indicates whether there are additional records stored in neighboring buckets that were moved due to overflow from the original bucket. If the overflow counter indicates that there are more records in the neighboring buckets, the search operation continues by reading the address stored in the OFRB to continue searching within the same segment.

**Membership count**: The membership count in ESH is a 1-byte value that stores the number of alien elements or elements that have been pushed to the bucket because their hashed bucket was already full. This count is maintained to keep track of how many elements have been moved to the bucket, which helps to facilitate searching operations.

In this scheme, every operation must first check the consistent state of the target bucket. When a running process wants to write or update the bucket, it acquires the lock and checks the overflow bit to determine if the bucket is full. If the conditions are met, the process proceeds with inserting a record into the bucket. However, if the bucket is already full, the process moves on to the next neighboring bucket, releasing the lock in the original bucket as described in Algorithm 1.

In the ESH scheme, each segment holds a fixed number of buckets consisting of 32 bytes of metadata and 224 bytes of key–value pairs. The directory entries point to these segments, which contain the buckets. The data are stored in the buckets as arrays of records, as depicted in Figure 4. A bucket serves as a storage unit for records generated by the hashing function, as well as records moved from neighboring buckets. During the insertion of records, if a bucket has more available space compared to its neighboring buckets, the records are moved to these neighboring buckets instead of triggering doubling. This approach is utilized when a particular bucket does not have sufficient space for insertion, but there are neighboring buckets that are free or have enough space to accommodate the records.

---

**Algorithm 1** ESH key–value insert algorithm

---

1: def ESH_insert(key,value):
2: hashval=hash(key);
3: overflow_counter ← 0
4: try:
5: #check consistency and get the lock to the targeted bucket
6: target_bucket=get_segment.bucket(hashval)
7: state = target_bucket.state
8: **if** state = 0 #check if the bucket is consistent **then**
9:    **if** get_segment.bucket=full **then**
10:       change overflow_bit=1
11:       target_bucket=get_segment.bucket← next
12:       increment overflow_counter & membership count
13:    **end if**
14:    Lock target_bucket
15:    **if** target_bucket.count<= segment_size **then**
16:       target_bucket.insert(key, value, hashval)
17:    **end if**
18:    **if** target_bucket.bucket != full **then**
19:       target_bucket.insert(key, value, hashval)
20:    **end if**
21:    target_bucket=get_segment.bucket(hashval)
22:    update membership_counter
23:    goto try:
24:    update overflow_record_address
25:    Unlock target_bucket
26:    return Insertion_result
27: **end if**

---

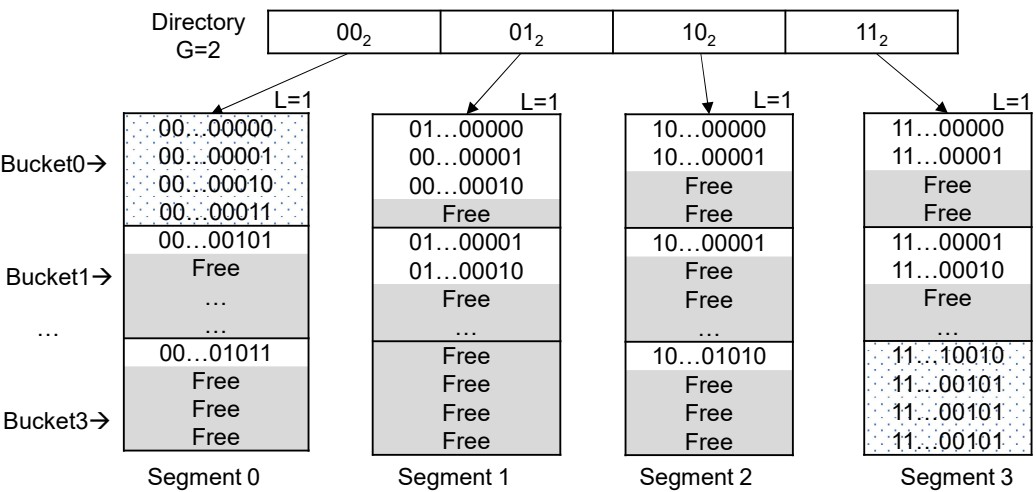

**Figure 4.** Segmented extendible hashing with cache-conscious feature.

To ensure concurrency control, the ESH scheme adopts the initial CCEH scheme, which utilizes locks and a flag to determine if a bucket is clean. This choice is based on the reasonable performance demonstrated by the CCEH scheme in the context of extendible hashing. When inserting records into the buckets, if there is no available space in any of the buckets within the same segment, the scheme checks for available space in the neighboring buckets. This approach is similar to how the CCEH scheme uses trackers for global and local depth during splitting and directory doubling operations.

To efficiently utilize the available buckets, particularly those that are partially filled or empty, the scheme employs a traversal process within the same segment to find the appro-

priate location for inserting a record in neighboring buckets. This traversal is constrained to remain within the segment to avoid complicating the lookup process. During each insertion, the record is hashed to the correct bucket, and if that bucket is already full, the scheme tracks other buckets with fewer elements and inserts the record using probing techniques. The records within the buckets are accessed as arrays, and each bucket contains a fixed number of elements, simplifying the search operation, as seen in Algorithm 2. If a value has been moved to another bucket, it can be accessed by looping through the records belonging to the same segment.

---

**Algorithm 2** ESH key–value search algorithm

```
 1: def ESH_search(key):
 2: #get the target_segment and target_bucket
 3: hashval=hash(key);
 4: try:
 5: target_segment = get_segment(hashval)
 6: #check current state
 7: target_bucket = get_bucket(hashval)
 8: if get_bucket(hashval) != target_bucket then
 9:     goto retry:
10:     # activate the bucket lock
11: end if
12: lock = target_bucket.state
13: if lock not set  then
14:     goto retry:
15:     search_result = target_bucket.search(key)
16:     if search_result is not in target_bucket.search(key) then
17:         # in case overflow was there, search in the segment.
18:         search_result = target_bucket ← next
19:         target_segment.target_bucket ← next
20:     end if
21:     search_result = target_bucket.search(key)
22: end if
23: if search_result is not NULL then
24:     return search_result;
25: else
26:     search_result = NULL
27: end if
28: return search_result
```

---

*4.2. Implementation*

Based on the properties of persistent memory and the utilization of hashing techniques, we have implemented a mechanism to minimize performance impact and optimize memory usage. Our approach focuses on reducing the frequency of full-table rehashing, which can significantly impact tail latency. In this, we aim to improve overall performance and maximize the efficiency of memory utilization.

The process of hashing records in this approach is similar to the process of hashing records to a bucket in CCEH for storage. When there is a request to insert a record into a bucket that is already full while the neighboring buckets have available space, the scheme checks if the insertion necessitates segment splitting. This ensures a balanced and comprehensive insertion across all buckets within the same segment before triggering a rehashing process. If there are neighboring buckets that are either empty or partially filled, it is more cost effective to store the records in those buckets rather than initiating directory doubling. This operation is facilitated by storing records in buckets and accessing them as arrays. Directory doubling and rehashing of records are only initiated when there is no space available in any of the buckets within the segment. The insertion process can

be identified by a pointer, which assists other operations and serves as an indication that certain records have been shifted to other buckets within the segment.

The deletion of records in this approach follows the overwrite method employed by the initial scheme, where invalid values can be overwritten with valid ones. In the case where a record is stored in a different bucket due to the bucket being full, the deletion operation will nullify the record in the respective bucket, thereby marking the space as free. For instance, if a record is originally hashed to bucket *b0* but is moved to bucket *b3* due to the bucket being full, the deletion process will first search for the record in bucket *b0*. If the record is not found, the search will continue in bucket *b3* by looping through the records in the bucket. Once the record is located in the intermediate buckets or in *b3*, it will be nullified, rendering it invalid. If a record is deleted from a filled bucket, the system will move the record from another bucket back to its original hashed bucket to facilitate future lookups, unless data access speed is a critical concern. However, this operation has lower priority, as the bucket size is equivalent to the size of the cache line. When all buckets are exhausted, a segment split will be performed to rehash all records into their appropriate buckets.

### 4.3. Concurrency

Developing a hashing scheme that efficiently scales on multicore machines utilizing persistent memory remains a complex task. ESH addresses this challenge by striving to create a scalable and efficient hashing scheme specifically designed for persistent memory. The goal is to achieve persistence without imposing excessive overhead, thereby contributing to the advancement of hashing schemes in the context of persistent memory.

### 4.4. Recovery

Failures can occur when a record is not fully written to the hash table. Instances like power failures or system crashes during the modification of the hash table can lead to such failures. For time-sensitive applications, it is crucial to have a recovery system that can swiftly recover from such failures and resume services. In ESH, records are stored as soon as they are received, ensuring that keys are also stored promptly. If a record is only partially written and the valid key values are not stored in the valid segment, it will be ignored. During startup, the hash directory is reconstructed following the principles of extendible hashing, and the directory and local depth are recovered by loading the records into memory. To evaluate the recovery performance, we conducted tests and comparisons with the previous scheme. We ran the hashing scheme for insertions, intentionally killed the process while it was loading records, and then restarted the scheme to measure the time it took to accept requests for various data sizes [18]. The results demonstrated that our scheme performed competitively in terms of recovery time. For instance, the recovery time required for our scheme to handle 1 million records was 103 ms, while it took 101.7 ms for CCEH. This result holds significance even for smaller data sizes, as the recovery time is nearly comparable to other schemes.

## 5. Experimental Evaluation and Results

In this section, we conducted an experimental setup to evaluate our scheme and compare it with state-of-the-art schemes that are designed for dynamic hashing on persistent memory. For this evaluation, we selected CCEH and Dash as the comparison schemes. Both of these schemes utilize the extendible hashing scheme, with CCEH emphasizing cache consciousness and Dash aiming to reduce read and write operations on a 256-byte bucket size.

The experimental results shows that our scheme:

- Efficiently utilizes the space at the segment level, ensuring that no empty space is left within a segment before triggering a segment split operation or directory doubling.
- In a multicore environment, our scheme exhibits strong scalability in terms of performance compared to state-of-the-art hashing schemes that employ similar techniques.

- Our scheme achieves a high load factor without compromising performance and recovery, all while maintaining minimal costs. It demonstrates competitive performance in this regard.

*5.1. Experimental Setup*

The experiments were conducted on a server equipped with an Intel Xeon® Gold 5218 CPU® running at 2.30 GHz. The server featured 32 cores (64 hyper threads) and cache sizes of 32 KB for both the data and instruction cache, 1024 KB for the L2 cache, and 22,528 KB for the L3 cache. It was also equipped with 256 GB of Optane DCPMM (2 × 128 GB) DIMMs in memory mode and 32 GB of DDR4 DRAM. The server ran Ubuntu server 18.04.4 LTS with kernel version 5.4.9-47-generic. The persistent memory development was facilitated using PMDK 1.8, and all the code was compiled using GCC 9.0 as indicated in Table 1.

**Table 1.** Experimental setup.

| Setup Used | Specifications |
|:---:|---:|
| Server | Intel Xeon®Gold 5218 |
| | Processor—Intel CPU® 2.30 GHz 32 cores (64 hyper threads) |
| | 32 GB DDR4 RAM |
| | 256 GB Optane DCPMM (2 × 128 GB) DIMMs |
| OS | Ubuntu server 18.04 LTS with Kernel 5.4.9-47-generic |
| Platform | PMDK 1.8 and GCC 9.0 |

Experiment parameters: These were carefully selected to ensure a fair comparison with prior works such as CCEH [14] and Dash [18], which also employ dynamic hashing on persistent memory (PM). To address data issues, threads were pinned to their respective physical cores in the multicore environment. Given that all the compared schemes utilize extendible hashing, similar setups were used to obtain reliable and comparable results. Hashing schemes like level hashing [15], which do not utilize the PM physical device for evaluation, were not considered in this scheme. Additionally, previous studies concluded that CCEH [14] outperformed these schemes in DRAM implementation. For a fair comparison, the experiment parameters were set accordingly. CCEH uses a segment size of 16 KB, a bucket size limited to 64 bytes, and a probing length of four. Dash, on the other hand, employs a bucket size of 256 bytes (equivalent to four cache lines) and a segment size of 16 KB. Therefore, in the case of ESH, we chose a bucket size of 256 bytes and a segment size of 1 KB, with a probing distance of four.

In prior work on CCEH, an increase in performance was observed as the segment size increased. This is attributed to the larger segment size allowing more data to be stored in memory and reducing the frequency of probing. In ESH, each bucket stores 28 key–value pairs of 8 bytes each, totaling 224 bytes, along with 32 bytes of metadata. A single segment accommodates four buckets. This configuration was chosen to effectively utilize the advantages of the Optane persistent memory buffer, which issues a 256-byte write block. Although the previous CCEH [14] scheme utilized a 64-byte bucket size to fit within a cache line, this was limited by the PMEM write block. By extending the bucket size to a 256-byte block, we achieve a balance where two blocks can be efficiently utilized. As the data size increases, the size of the hashing table also increases, resulting in a higher number of directory entries and segments following the extendible hashing technique.

The performance and effectiveness of the hashing scheme were assessed through a series of benchmarks designed to simulate realistic workloads. These benchmarks were used to test the scheme's performance under specific loads and collect diagnostic information to identify and address potential bottlenecks. The benchmarking tools employed evaluated various hashing operations, including insertion, search, and update operations. By comparing the results of these benchmarks with those of other schemes, the scheme's performance could be assessed and its competitiveness evaluated.

To conduct the experiment, an initial data load of 10 million records was pre-loaded using the GCC file that defines Hash_bytes. This file utilizes the MurmurHashUnaligned primitive [54], which is a public domain hash function known for its fast and high-quality hashing capabilities. This approach ensured a fair comparison with Dash, which also utilized a hashing index. To thoroughly test the performance and robustness of our hashing scheme, we employed both uniform and skewed/zipfian distributions with varying sizes. These distributions were used to stress test the scheme and assess its performance under different scenarios.

The experimental results demonstrated that our scheme outperformed others, primarily due to the cross-bucket record storage technique that reduced the frequency of costly directory doubling operations. Additionally, the metadata incorporated in our scheme helped minimize unnecessary operations and contentions, further enhancing its performance. For the experiments, a fixed-size key–value structure consisting of 8-byte integers was used. Furthermore, the experiment included variable-length keys generated by Dash [18] using their benchmark's pre-generated variable keys, ensuring consistency and comparability with previous studies.

### 5.2. Comparative Performance

In order to assess the fundamental performance of our design, we conducted measurements on single-threaded operations using different data sizes. We specifically focused on data insertion operations, as they provide insights into the underlying performance limitations. By comparing the performance of these operations, we were able to evaluate the efficiency of our scheme.

The results, depicted in Figure 5, clearly demonstrate that ESH outperformed both the CCEH and Dash hashing schemes. In fact, our scheme exhibited a 30% improvement compared to CCEH and a 4% improvement compared to Dash when inserting large datasets. These findings highlight the superior performance of our scheme in handling larger volumes of data.

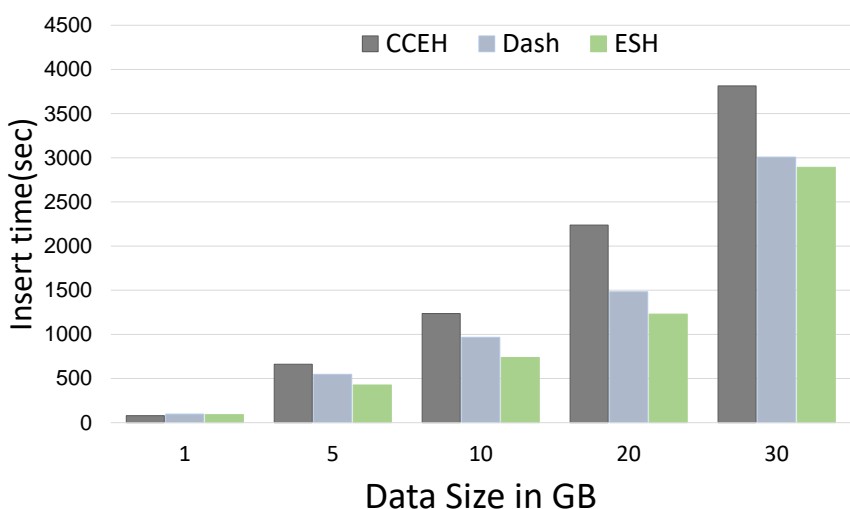

**Figure 5.** Single-thread data insertion performance comparison under a fixed key length.

As depicted in Figure 5, ESH consistently outperformed the other schemes in terms of performance. This can be attributed to the effective utilization of time that would otherwise be spent on the expensive operation of directory doubling. By delaying and optimizing the directory doubling process, ESH is able to allocate that time to insert records into the already created hashing table. This approach significantly enhances the overall performance of the scheme, resulting in improved efficiency compared to the other schemes. For a large amount of data, the overhead of full-table rehashing is significantly higher, but in this mechanism we managed to delay and add more records to the already created buckets. This

is because rehashing is faster as the number of buckets is small, and it is not advantageous to store records in other buckets than the designated buckets in the event of overflow.

The experiment in Figure 5 alone is not detailed enough to show how ESH outperforms the schemes. On top of this experiment, it was critical to show the number of probes in the event of an overflow, since probing is used to traverse the other buckets before we find a free slot to insert the key and then save the address in the OFRB. The number of buckets and segment sizes should be equal for all approaches for a fair analysis, therefore, there should be an equal number of overflows during key insertions. So it is more interesting to see the total number of probes of ESH as compared to Dash and CCEH. The insert time alone does not show the inner details of why ESH outperforms the other solutions.

When examining the performance of Dash and ESH with increasing sizes of key–value stores, both schemes demonstrated comparable levels of performance. However, they both outperformed CCEH, with ESH achieving up to three times better performance. This can be attributed to the utilization of metadata in Dash and the use of fingerprints in ESH, which contribute to more efficient operations and improved performance compared to the CCEH scheme.

While CCEH initially exhibited better performance with small data sizes, its performance gradually decreased as the data size increased. This can be attributed to its use of cache-line-sized data, which allows for faster flushing and completion of hashing operations within a shorter timeframe compared to other hashing schemes. However, as the data size grows, the limitations of this approach become more apparent, resulting in reduced performance compared to alternative schemes.

Our scheme treats the insertion of a record to allocated positions, as well as cases of overflows. We will show the hash table operation as follows. When an insert operation is requested for a record, if there is no space available in the original bucket, but there is space available in a neighboring bucket, ESH stores the record in the neighboring bucket to delay the rehashing. This insert operation is illustrated in Figure 6, as well as how it is handled in ESH as a result. As a result, when a thread attempts to insert a record into bucket (①), it checks the state of bucket 0 to see if it is consistent. If the state is consistent, it examines the overflow bit (OB) to see if there is available space or not in bucket 0. In this scenario, however, the initial bucket (bucket 0) is already full. As a result, the thread obtains the adjacent bucket (bucket 1) in the same segment and searches for vacant space in bucket (②). In addition, bucket 1 is already full in this scenario. As a result, the thread continues to look for empty space in the next neighboring bucket (bucket 2) (③). Bucket 2 has free space in this situation. Thus, the thread holds bucket 2's lock and puts the record into bucket 2 (④). Then, it updates the neighboring bucket address of its own bucket (bucket 0) with bucket 2. It also adds one to the overflow count of bucket 0 and one to the member count of bucket 2. Finally, it determines whether or not the member count of bucket 2 is maximum (full). If it is full, bucket 2's overflow bit is set to 1. In this way, ESH can maximize the utilization of more free space in buckets in a segment and postpone the rehashing operation.

Searching for a record that has been relocated to another bucket can be accomplished by looping through records from the same segment, which has a lower overhead than doubling operations.

Figure 7 illustrates the search and delete operations in detail. When a thread initiates a search for a record, it begins by searching the original bucket where the record is initially hashed (i.e., bucket 0). The thread first checks the consistency of bucket 0. If the state is consistent, the thread proceeds to check the overflow bit (OB). If the overflow bit is 0, it indicates that the key is located in the original bucket. As a result, the thread can easily retrieve the corresponding value from the key in the original bucket. On the other hand, if the overflow bit is 1, it implies that the key could be present in either the original bucket or neighboring buckets. In this scenario, if the key is not found in the original bucket, the thread moves on to search in the neighboring buckets. As depicted in the figure, we can observe that the original bucket (bucket 0) does not contain the desired key (①).

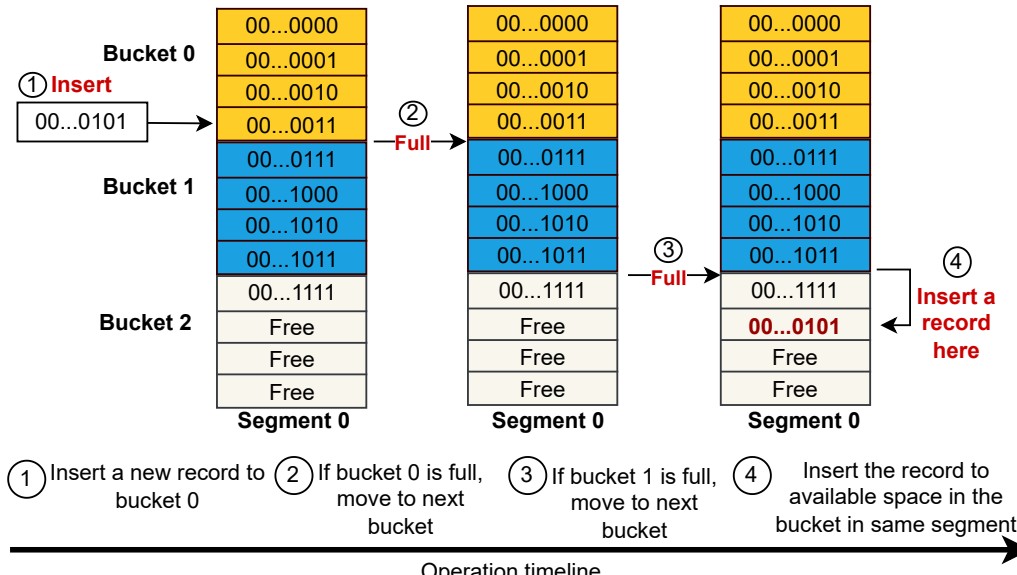

**Figure 6.** Record insertion operation in ESH.

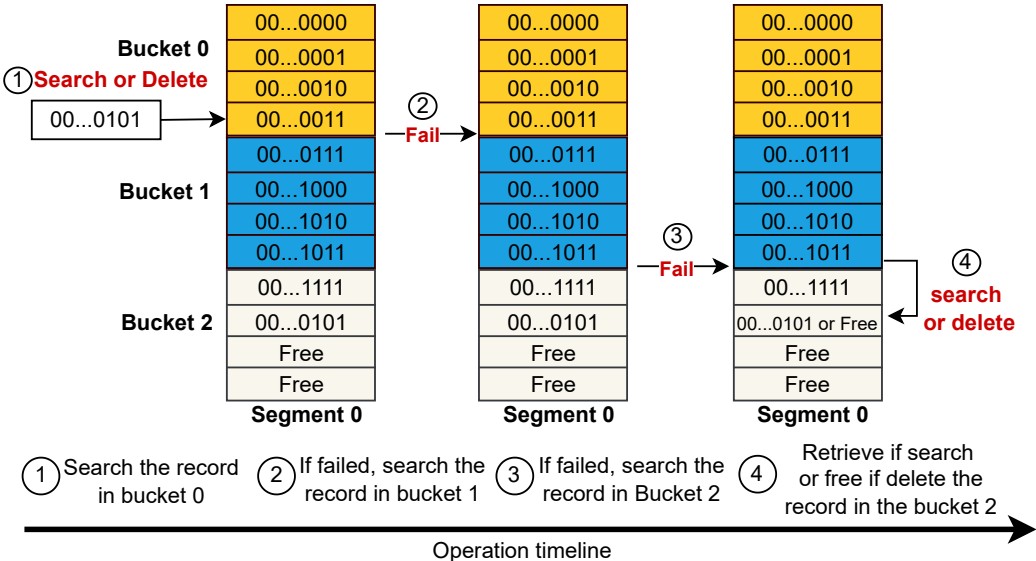

**Figure 7.** Record search or deletion operation in ESH.

Consequently, the thread proceeds to search for the key in the neighboring bucket (bucket 1) within segment (②). However, even in bucket 1, the key is not found. Hence, the thread continues its search in the next neighboring bucket (bucket 2), as denoted by (③). Eventually, the key is found in bucket 2, allowing the thread to retrieve the corresponding value from it.

In the case of a delete operation, the thread follows a similar process as the search operation described above. If the thread successfully locates the key, it proceeds to delete the corresponding record (④). This deletion operation involves reducing the overflow count of bucket 0 by 1 and decreasing the member count of bucket 2 by 1. Additionally, if bucket 2 is not already full, the overflow bit of bucket 2 is updated to 0.

From the aforementioned explanation, we can conclude that the figure clearly demonstrates the sequence of search and delete operations in the given system. As records are continuously received from the client, our approach allows for their insertion into the buckets as long as there is available space in the segment. In fact, our strategy ensures that at least one record is stored in the neighboring bucket before invoking the segment split or

directory doubling operation. This unique advantage becomes even more pronounced as the amount of data continues to grow. Consequently, the delay in the split and directory doubling operations becomes more significant, further enhancing the overall efficiency of our mechanism. From the experimental results, our scheme has shown that there is a delay in directory doubling as there are records stored in neighboring buckets after the initial bucket fills up. During the experiment, the delay became noticeable after the segment splitting and directory doubling happened in the first seven rounds, where more records are inserted before a directory doubling occurs.

Unlike other schemes, where it is a must for the scheme to trigger a full-table rehashing operation even if neighboring buckets are empty, ESH employs an effective approach where it delays this expensive operation until all empty spaces are exhausted. In ESH, when a bucket becomes full, we utilize the neighboring buckets within the same segment by moving records into them. This strategy delays the need for a full-table rehashing operation, ensuring efficient utilization of the hash table. As a result, ESH optimizes performance by avoiding unnecessary rehashing when the hash table is actually only partially filled.

By increasing the size of the buckets in our design, we were able to accommodate a larger number of records in the hash table that matches the access block size of the persistent memory. This approach leveraged the benefit of the persistent memory write block, allowing us to store more records within each bucket. As a result, we achieved improved insertion capacity and reduced the frequency of full-table rehashing. Additionally, this optimization helped to minimize unnecessary directory doubling, which can be a costly operation impacting the overall performance of the hashing scheme.

To ensure successful insertion, the hashing function accesses the segments by utilizing the directory entry that provides pointers to the corresponding segments. In this design, each segment is composed of four buckets, which can be accessed through the segment's entry pointer. During the insertion operation, the segment is checked to determine if any of its buckets have sufficient space to accommodate the newly inserted value. If a bucket within the segment is full, the scheme initiates a bucket-splitting process and inserts the new value into the appropriate location within the split buckets.

This process continues until all the buckets in the segment are filled. If the target bucket is already full but there are other buckets within the same segment that still have available space, the metadata of each neighboring bucket is examined. In this case, the new value is moved to one of these buckets, and the original bucket's OB and OFRB pointer are modified for future access. By utilizing the available space in the buckets within the segment, this approach effectively fills up the segment before triggering the directory doubling process, making efficient use of all the bucket spaces.

Our scheme efficiently inserts records into all the buckets within the same segment, making optimal use of the hash table before initiating the costly full-table rehashing process. To accomplish this, we examine the header information of each bucket to determine the available free space and store the new record in the buckets with fewer key–value stores already present.

The process of directory doubling involves the rehashing and redistribution of all records to buckets. During a search operation, it starts from the target bucket. If the search key does not match the header information, there is no need to loop through the elements in the bucket. Instead, it moves on to the next neighboring bucket. This transition to the next bucket is only initiated if the "OB" flag indicates that the bucket contains overflowed elements that have been moved to the neighboring bucket. Otherwise, the search concludes within the segment, and the result indicates that no element was found. However, if there is an overflow indicator *OC* for a particular bucket, the search operation will proceed to the neighboring bucket and examine the header or metadata of each bucket to see if the search key matches any elements that have been moved from another bucket. These moved elements are identified by the *membership count* in the metadata. If the item is not found in the initially hashed bucket and the neighboring buckets within the same segment, the search for the record will be unsuccessful.

As depicted in Figure 8, the search performance of ESH outperforms CCEH by a factor of two and demonstrates a 10% improvement compared to Dash. This achievement can be attributed to the utilization of metadata information, which facilitates the search operation in examining all buckets within the same segment for the desired element. Starting from a specific bucket, the search checks the bucket's header to determine if the value exists. If the value is not found in the bucket and there is no overflow indicator $OFRB$, the search concludes within the same bucket. However, if an overflow $OFRB$ indicator is present, the search continues from the address pointed to by the $OFRB$ within the same segment. The experimental results clearly demonstrate the effectiveness of this mechanism and the utilization of bucket-level headers.

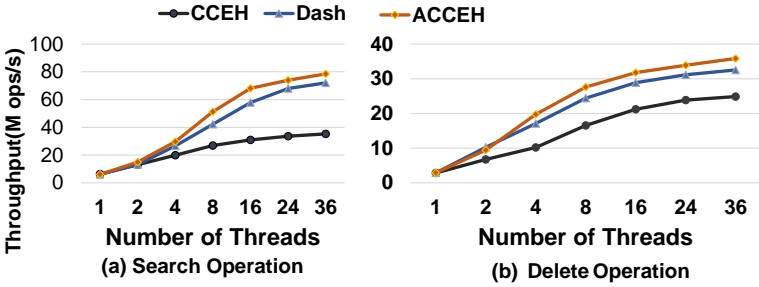

**Figure 8.** Throughput Comparison for a varying number of threads.

Furthermore, in terms of delete operations, ESH exhibited superior performance compared to both schemes. The experimental results indicate that ESH outperformed CCEH and Dash by 10% and 3.2%, respectively. This performance gain in delete operations can be attributed to the fact that it builds upon the search operation, thus contributing to an overall improvement in performance.

### 5.3. Benefits of Metadata

The enhanced performance and scalability of ESH can be attributed to its efficient utilization of metadata or headers at the bucket entry, which significantly reduces data access. This advantage is evident when comparing ESH with other state-of-the-art schemes that employ similar hashing schemes across a range of data sizes and varying numbers of threads. The improved throughput observed in Section 5.4 for insertion, search, and delete operations, with different thread configurations, underscores the benefits of using metadata to streamline the search process. By accessing bucket information from the metadata, unnecessary time spent searching records within the bucket is minimized, resulting in reduced PM access and improved response times. The insertion operation leverages the $OB$ and $OFRB$ metadata elements to determine the bucket status, while delete operations also consider the $OC$ metadata before proceeding to access the records within the bucket.

### 5.4. Concurrency

In a multi-threaded environment, performing multiple queries can pose challenges due to concurrent access to a hash table. This is particularly true for costly operations like full-table rehashing, which require exclusive access to the entire hash table and can potentially block subsequent operations, leading to increased response times. This challenge becomes more critical as the size of the hash table increases. To address this, we conducted an evaluation of the latency of concurrent operations, specifically, insertion and search operations.

When running a large number of insertions in a multi-threaded environment, the insertion throughput of all the compared schemes increased. Both Dash and ESH exhibited superior overall performance in this scenario. ESH, in particular, demonstrated better performance by reducing lock contention to the single-bucket level, whereas Dash locks a segment for operations. This reduction in lock contention contributes to the improved performance of ESH in concurrent operations.

### 5.5. Scalability

Figure 9 illustrates the scalability of the insertion operation for different hashing schemes with varying numbers of threads. In today's world, with ever-growing amounts of data, the scalability of a system holds paramount importance. Our scheme has demonstrated its scalability across a spectrum of thread counts, consistently outperforming other schemes tailored for the same architecture. As the size of the hashing key increases, ESH exhibits superior scaling compared to the other schemes. Specifically, ESH demonstrates 34% better scalability than CCEH and 12% better scalability than Dash in terms of insertion operations. In the case of search operations, Dash and ESH show near-linear performance, while CCEH lags behind due to the impact of locking and cache-line-level hashing, which is affected by the size of the hash table and requires a larger number of persistent memory writes.

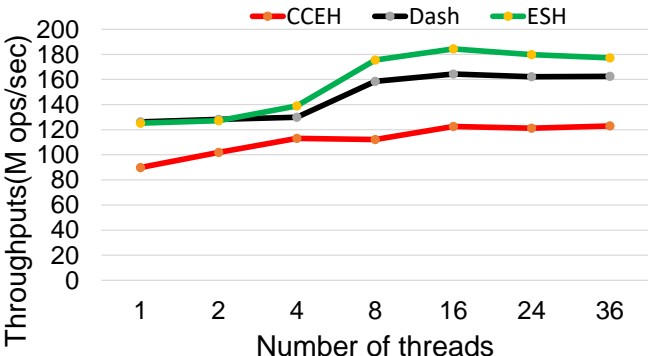

**Figure 9.** Multiple thread performance comparison.

Although all three schemes employ similar extendible hashing techniques, the differences lie in the design of the lock strips [55], which allows locking to fit into the CPU cache, and the impact of the building blocks of persistent memory on log writing and block flushing [56], which affects write bandwidth. ESH demonstrates higher performance in insertion, search, and delete operations and scales effectively with varying numbers of threads. This is attributed to the modifications made in ESH, such as delaying full-table rehashing by making full use of the allocated hash table. Furthermore, the implementation of optimistic locking at the bucket level allows other threads to access different buckets, reducing the response time for each thread.

### 5.6. Load Factor

By limiting the number of buckets within a single segment to four, the total number of buckets in a segment is minimized. This limitation leads to a better load factor, which represents the ratio of the number of elements stored in each bucket to the total number of positions available in the segment. A higher load factor indicates improved memory efficiency in the hash table. Since operations like insertion and search are performed at the segment level, having a larger number of buckets within a segment does not offer an advantage. In fact, limiting the number of buckets in a segment results in better linear probing performance. ESH achieves this by employing a balanced insertion strategy, where records are moved to neighboring buckets to improve the load factor.

However, increasing the segment size or bucket size reduces the directory size at the expense of a decreased load factor. In order to ensure efficient storage of records within the hash table, ESH prioritizes achieving a higher load factor by utilizing a limited number of buckets within each segment.

Figure 10 depicts the variations in load factor as the number of insertions increases, measuring the load factor at different record counts in the hash tables. CCEH demonstrates a relatively stable result, as it splits a segment after four cache line probes. This finding aligns with observations made in [18], where it is noted that longer probing lengths increase

the load factor at the expense of performance, while shorter probing lengths result in premature splits.

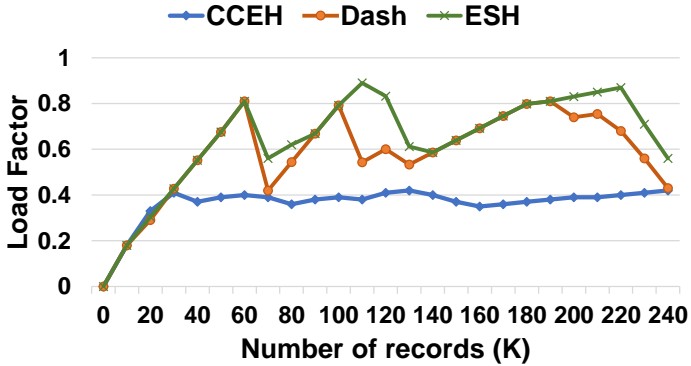

**Figure 10.** Load factor with respect to the number of items inserted into the hash table.

In contrast, both Dash and ESH exhibit higher load factors, indicating that the design changes have proven effective. The modifications in Dash and ESH have led to improved load factors compared to CCEH. The improved load factor in ESH compared to Dash can be attributed to the delayed segment split and directory doubling operation. By storing records in all available spaces of neighboring buckets before initiating the split or doubling operation, ESH effectively utilizes the existing space. The experimental results exhibit "valley" shapes, which correspond to the points where segment splitting or directory doubling occurs, leading to rehashing.

As a result of these design choices, ESH achieves a load factor of up to 91%, surpassing the load factors achieved by state-of-the-art schemes.

### 5.7. Recovery

Recovery is an essential feature of a system that aims to restore data to its normal state in the event of failures. System failures can occur for various reasons, such as power outages or unknown causes, leading to service unavailability or slowdown. In the case of a persistent hash table, similar to Dash, we employed a comparable testing method to ensure a fair comparison. This involved loading a set of records for a certain duration and then intentionally terminating the process responsible for the loading operation. Subsequently, we measured the time taken by each scheme to resume accepting incoming requests.

The results presented in Table 2 demonstrate that both ESH and Dash exhibit rapid recovery times, even for large amounts of data, while CCEH requires more time compared to its competitors and exhibits scalability issues as the amount of data increases. The swift recovery times of Dash and ESH can be attributed to their utilization of specific markers: Dash employs a clean marker, whereas ESH employs a "state" marker. These markers serve to check whether the system was cleanly shutdown in a previous operation. During the recovery process, the system reads these markers and restores the persistent memory to a consistent state. It is worth noting that ESH, which operates on a per-bucket basis, requires slightly more time for recovery compared to Dash, which is negligible even for large record sets. Additionally, both schemes address other consistency concerns during multi-threaded operations through the inclusion of a "lock" signal in the metadata.

**Table 2.** Recovery time (ms) comparison with respect to data size.

| Hashing Schemes | Record Size in GB | | | | |
|---|---|---|---|---|---|
| | 1 | 5 | 10 | 20 | 30 |
| CCEH | 50 | 121 | 256 | 503 | 1082 |
| Dash | 62 | 63 | 65 | 65 | 65 |
| ESH | 62 | 65 | 66 | 66 | 67 |

## 6. Conclusions

Optimal memory utilization is indispensable across all applications, especially in those demanding high memory usage, where it becomes imperative. Given the evolving trend in memory development, storage systems must adapt to these new architectures. In this research, a pioneering approach is presented that maximizes the use of unoccupied space in adjacent storage units for record storage. This approach significantly lowers PM access and optimizes bandwidth consumption. By efficiently making use of available capacity in nearby nodes, redundant directory doubling procedures are postponed, resulting in a more effective deployment of persistent memory space. Additionally, scalability tests have demonstrated that this scheme performs admirably in multi-threaded environments. This strategy enables us to efficiently harness the persistent memory capacity. To enhance performance, we put forward a minimal locking approach where a bucket is locked only during a write request, enabling concurrent access/read operations on buckets within the segments by other processes. This facilitates multiple access points, leading to increased scalability in multi-threaded environments. Furthermore, our scheme exhibits superior performance compared to an alternative approach for both uniform and skewed data distributions. Our approach exhibits superior performance compared to current leading methods, regardless of whether the data distribution is uniform or skewed. As a result, it presents an alternative solution to enhance CCEH by mitigating the overhead linked with split management and deferring the resource-intensive directory doubling operation in systems employing an extendible hashing scheme on persistent memory.

In the future, we intend to assess the scheme across a diverse array of applications and various workloads in real-world scenarios to further refine it for use at a production level. Additionally, we envisage incorporating additional tests and seeking peer evaluations to establish a comprehensive framework adept at handling hashing for a wide spectrum of applications, based on the insights gleaned from these evaluations.

**Author Contributions:** Conceptualization and methodology, D.R. and H.Y.Y.; methodology, D.R. and H.Y.Y.; experimental activity, data processing, and validation D.R.; writing original draft preparation, review and editing, D.R., H.Y.Y. and J.H.; supervision and funding acquisition, D.R. and J.H. All authors have read and agreed to the published version of the manuscript.

**Funding:** This work is partially funded by the National Research Foundation of Korea (NRF) grant funded by the Korean government through BK21 Stage 4 Graduate School Innovation Education Research Group (No. 3789). (Corresponding Author: Dereje Regassa).

**Institutional Review Board Statement:** Not applicable.

**Informed Consent Statement:** Not applicable.

**Data Availability Statement:** Avialable on request.

**Conflicts of Interest:** The authors declare no conflict of interest.

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
