# Peer review of "ESH: Design and Implementation of an Optimal Hashing Scheme for Persistent Memory"

_applsci, doi:10.3390/app132011528_

Round 1

Reviewer 1 Report

This paper claims to propose an Efficient and Scalable hashing scheme that significantly improves memory efficiency, scalability, and overall performance on persistent memory.

-Abstract needs to include the major findings.

-Introduction should be revised: (1) Introduce the problem (2)discuss about some of the existing research works (3)identify the gap or scope of improvement (4) discuss in order to address the identified gaps what is the methodology used (5) list out the contributions (6) Organization of paper.

-What is the need of unnecessary background in the Second section?

-Figure 3. Overall architecture of bucket and Segment structure for Scalable hashing scheme    needs to be explained in detail

-4.1 experimental setup can be represented as  a table. Also, justification of the parameter values is necessary.

-Graph quality should be improved considering the uniform size of text and figures.

-Results are not analyzed in detail

-I can find content with repetition of the same thing in different way. Section 4 can be improved by including only solid content.

-Related work should include some recent works, such as:

A Design Space Exploration and Evaluation for Main-Memory Hash Joins in Storage Class Memory

Comparative study on hash functions for lightweight blockchain in Internet of Things (IoT)

MixNeRF: Memory Efficient NeRF with Feature Mixed-up Hash Table

-Conclusion is poorly written. What are the use cases and limitations of the approach?

Pl proofread the paper for typos and grammar issues.

Reviewer 2 Report

The article titled "ESH: A Design and Implementation of an Optimal Hashing Scheme for Persistent Memory", is well written and contributed by the authors. The present version of the article can be accepted for publication. 

The article titled "ESH: A Design and Implementation of an Optimal Hashing Scheme for Persistent Memory", is well written and contributed by the authors. The present version of the article can be accepted for publication. Some proofreading and grammatical corrections are required. 

Reviewer 3 Report

This manuscript introduces a new optimal hashing scheme for persistent memory. During my assessment, I have identified some minor concerns that need to be addressed:

1.  The Introduction section requires a comprehensive revision to incorporate a review of more recently published works in the field. By including recent research findings, the authors can establish the context of their study and highlight the novel aspects that differentiate their work from previous contributions.

2.   I find it somewhat unexpected to encounter the Related Work section placed towards the conclusion of this manuscript. Ideally, this section should follow immediately after the introduction or can be merged into the introduction section itself.

3.    The conclusion section requires rewriting to review the achievements made in the research and outline the future scope. The current conclusion lacks a thorough analysis of the results and their implications. The authors should repeat this section to provide a more comprehensive overview of the accomplishments and discuss potential future directions for research in the field.

Reviewer 4 Report

Manuscript is well organised, scitfically presented data. Can be accepted in current sate for publication

Accpet in current state
